

# The miRNA biogenesis in marine bivalves

Umberto Rosani[1], Alberto Pallavicini[2] and Paola Venier[1]

[1] Department of Biology, University of Padova, Padova, Italy
[2] Department of Life Sciences, University of Trieste, Trieste, Italy

## ABSTRACT

Small non-coding RNAs include powerful regulators of gene expression, transposon mobility and virus activity. Among the various categories, mature microRNAs (miRNAs) guide the translational repression and decay of several targeted mRNAs. The biogenesis of miRNAs depends on few gene products, essentially conserved from basal to higher metazoans, whose protein domains allow specific interactions with dsRNA. Here, we report the identification of key genes responsible of the miRNA biogenesis in 32 bivalves, with particular attention to the aquaculture species *Mytilus galloprovincialis* and *Crassostrea gigas*. In detail, we have identified and phylogenetically compared eight evolutionary conserved proteins: DROSHA, DGCR8, EXP5, RAN, DICER TARBP2, AGO and PIWI. In mussels, we recognized several other proteins participating in the miRNA biogenesis or in the subsequent RNA silencing. According to digital expression analysis, these genes display low and not inducible expression levels in adult mussels and oysters whereas they are considerably expressed during development. As miRNAs play an important role also in the antiviral responses, knowledge on their production and regulative effects can shed light on essential molecular processes and provide new hints for disease prevention in bivalves.

Corresponding author
Paola Venier, paola.venier@unipd.it

## INTRODUCTION

Different types of non-coding RNAs (ncRNAs) have gained attention for their powerful regulatory action on eukaryotic genes and other genetic elements (*Carninci et al., 2005*; *Mortimer, Kidwell & Doudna, 2014*). The process known as RNA interference (RNAi) exemplifies an evolutionary conserved mechanism of gene silencing based on small guide RNAs and specific interacting proteins (*Tomoyasu et al., 2008*; *Gammon & Mello, 2015*). Silencing RNAs (siRNAs) and microRNAs (miRNAs) take part to the same control machinery whereas Piwi-interacting RNAs (piRNAs) peculiarly silence germ-line transposons, among other roles (*Théron et al., 2014*; *Iwasaki, Siomi & Siomi, 2015*). Long noncoding RNAs (lncRNAs) in their normal or mutated forms can widely influence physiological and pathological processes, as multiple lines of evidence indicate their involvement in chromosome inactivation and epigenetic modifications, control of mRNA decay and translation, and DNA sequestration of transcription factors (*Huarte, 2015*; *Ruan, 2015*). More recently, circular RNAs have been identified as a group of competing

endogenous RNAs whose effects in the miRNA function and transcriptional/post-transcriptional regulation are now matter of study (*Qu et al., 2015*).

miRNAs are single-stranded RNA molecules of around 22 nucleotides, presenting conserved structural features and able to modulate the expression of eukaryotic genes by inhibition of mRNA translation or enhancement of mRNA decay (*Ambros, 2003*; *Bartel, 2004*; *Tarver, Donoghue & Peterson, 2012*). Up to now, diversified sets of miRNAs have been detected in five eukaryotic taxa (eumetazoans, silicisponges, vascular plants, *Clamydomonas* and *Ectocarpus* spp.) while they are apparently absent in protists (*Grimson et al., 2008*; *Tarver et al., 2015*). Depending on the annotation procedure, the number of human miRNAs varies from 523 to 1,881 miRNA precursors, as reported in MirGeneDB (*Fromm et al., 2015*) or in miRBase v. 21 (Kozomara, Griffiths & Jones, 2014), respectively. Overall, human miRNAs could target 30–60% of the transcribed genes (*John et al., 2004*; *Sand et al., 2012*), with implications in cell differentiation (*Berezikov et al., 2005*), cell death (*Xu et al., 2015*), stress responses (*Mendell & Olson, 2012*) and diseases (*Huang et al., 2014*; *Min & Chan, 2015*).

The miRNA biogenesis starts from pri-miRNA transcripts, mostly generated from RNA polymerase II in form of long non-coding RNAs and able to form a hairpin subsequently recognized by the so called microprocessor complex. DROSHA, a double-stranded RNA-specific ribonuclease III, and the RNA binding protein *Di-George syndrome Critical Region gene 8* (DGCR8) are the microprocessor's core proteins which allow interactions with the DDX5 helicase, the RNA binding protein Lin-28 and hnRNP A1, among other elements (*Jean-Philippe, Paz & Caputi, 2013*; *Hong et al., 2013*). During the recognition of pri-miRNAs at the dsRNA-ssRNA junction, DGCR8 acts as a crucial molecular anchor and directs DROSHA to cleave 11 bp away from the junction, with consequent release of hairpin-shaped pre-miRNAs (*Denli et al., 2004*). Pre-miRNAs are firstly exported to the cytoplasm via the *Exportin5* (XPO5) by interaction with the small GTPase RAN; then, they are further processed by the RISC loading complex, composed by the endoribonuclease DICER, the RNA binding protein TARBP2 and Argonaute proteins (*MacRae et al., 2008*; *Miyoshi et al., 2009*). The evolutionary conserved Argonaute proteins are specialized in binding small RNAs and exist in several isoforms, with AGO and PIWI representing two distinct subclades (*Tolia & JoshuaTor, 2007*; *Ender & Meister, 2010*).

AGOs select the 'guide' miRNA strand necessary for targeted gene silencing and, therefore, are responsible for final miRNA maturation. Several other proteins have been demonstrated to cooperate in miRNA processing and functions (*Ender & Meister, 2010*). In fact, AGOs operate transcriptional repression and cause mRNA decay by interacting with the GW-rich N-terminal region of GW182, a protein associated with cellular P-bodies (*Van Kouwenhove, Kedde & Agami, 2011*). Other proteins involved in the mRNA turnover (CAF1, PABPC1, eIF4G; CCR4-NOT and PAN2-PAN3 deadenylation complexes; in human somatic cells, also the decapping complex DCP1-DCP2 and at least four helicases, DDX5, DDX6, DDX17 and DDX42) may cooperate with the AGO-GW182 complex to reduce the mRNA translation efficiency (*Nottrott, Simard & Richter, 2006*; *Fabian & Sonenberg, 2012*).

Unlike AGOs, the PIWI proteins specifically interact with piRNAs to participate in the germline specification, gametogenesis, transposon silencing and in the maintenance of genome integrity (*Carmell et al., 2007*; *Malone & Hannon, 2009*; *Ghildiyal & Zamore, 2009*; *Siomi et al., 2011*). The piRNA mechanism of action is not so well defined but probably it involves the *arginine methyl-transferase* PRMT5, *tudor domain-containing proteins* (TDRDs) and the *Maelstrom* protein (MAEL) (*Sokolova et al., 2011*).

With the widespread and cost-effective use of Next Generation Sequencing (NGS) technologies, miRNAs have been deeply explored in non-model organisms, including bacteria (*Xu et al., 2014*), plants (*Rhee, Chae & Kim, 2015*) and viruses (*Kincaid & Sullivan, 2012*; *Diebel et al., 2015*). The basic set of genes involved in the miRNA biogenesis, and related protein interactions, are well known in mammals (*Lau & MacRae, 2009*), and also in other metazoans like *Cnidaria* (*Moran et al., 2013*), *Platyhelminthes* (*Resch & Palakodeti, 2012*) and insects (*Lucas & Raikhel, 2013*; *Hussain & Asgari, 2014*). Regarding mollusks, lists of miRNAs have been reported for a few species (*Jiao et al., 2014*; *Chen et al., 2014*; *Martín-Gómez et al., 2014*; *Zhou et al., 2014*), miRNA families have been investigated in the limpet genome (*Kenny et al., 2015*) and one study has considered bivalve DICER sequences for phylogenetic analysis (*Gao et al., 2014*). A general overview on the bivalve miRNA biogenesis complements is still lacking, so we took advantage of several genomic and transcriptomic datasets available for *Lophotrochozoa* (*GIGA Community of Scientists, 2014*) to identify and characterize the core elements involved in the miRNA formation pathway in *Mytilus* and *Crassostrea spp.* and other bivalves.

## MATERIALS & METHODS

Sequences coding for proteins centrally involved in the miRNA pathway, namely DROSHA, DGCR8, XPO5, RAN, DICER, TARBP2, AGO and PIWI, have been methodically identified in the genomes and transcriptomes of *M. galloprovincialis* (Mg) and *C. gigas* (Cg) as well as in other bivalve and non-bivalve species (66 species, listed in Table 1).

### Sequence retrieval and analysis

The Mg WGS project (ID APJB000000000.1 (*Nguyen, Hayes & Ingram, 2014*)) and the Cg genome draft (GCA_000297895 (*Zhang et al., 2012*)) were retrieved from GenBank, whereas the oyster genome annotations were obtained from Ensembl Metazoa release 29 (http://metazoa.ensembl.org/Crassostrea_gigas/Info/Index). A Mg reference transcriptome was produced using 18,788 ESTs of mixed tissues previously obtained by Sanger sequencing (*Venier et al., 2009*) and 453 million reads obtained by paired-end (2 × 100 bp) Illumina Hiseq2000 sequencing of digestive gland from North Adriatic Sea mussels (ID: PRJNA88481) (*Gerdol et al., 2014*), and from haemocytes, gills, mantle and muscle of Spanish mussels (ID: SRP033481) (*Moreira et al., 2015*). The quality of the sequencing readout was evaluated by the FastQC suite (http://www.bioinformatics.babraham.ac.uk/projects/fastqc/) discarding the reads with PHRED quality below 20 and presenting more than two ambiguous nucleotides. De-novo assembly was performed with Trinity, release 2013-08-14 (*Grabherr et al., 2011*), setting the minimum contig length at 200 bp and using default settings. Subsequently,
**Table 1  Organisms included in the present work.** Phylum, organism name, sequence origin and reference, ID used in phylogenetic trees and identified sequences are reported. Protostomia (green), Deuterostomia (orange) and novel protein sequences (numbers in bold) are well discernible.

| Phylum | Species | Sequence origin | Ref | Used ID | DROSHA | DGCR8 | XPO5 | RAN | DICER | TARBP2 | AGO or PIWI |
|---|---|---|---|---|---|---|---|---|---|---|---|
| Ctenophora | *Pleurobrachia bachei* | G | EM | Ple_bac | No | No | 1 | No | 1 | No | 4 |
| | *Mnemiopsis leidyi* | G | EM | Mne_lei | No | No | 1 | 1 | 1 | No | 4 |
| Porifera | *Amphimedon queenslandica* | G | EM | Aq | No | 1 | 1 | 1 | 2 | No | 2 |
| Placozoa | *Trichoplax adhaerens* | G | EM | Tri_adh | No | No | 1 | 1 | 2 | No | 1 |
| | *Nematostella vectensis* | G | EM | Nvec | 1 | 1 | 1 | 1 | 2 | No | 4 |
| Cnidaria | *Porites australiensis* | T | TSA | Por_aus | 1 | 1 | 1 | 1 | 1 | No | 6 |
| | *Anthopleura elegantissima* | T | TSA | Ant_ele | 1 | 1 | 1 | 1 | 1 | No | 6 |
| Nematoda | *Caenorhabditis elegans* | G | M | Ce | 1 | 1 | | 1 | 1 | 1 | 23 |
| | *Daphnia pulex* | G | EM | Dap_pul | 1 | 1 | 1 | 1 | 2 | 1 | 10 |
| | *Culex quinquefasciatus* | G | EM | Cq | 1 | 1 | 1 | 1 | 2 | 1 | 4 |
| | *Drosophila melanogaster* | G | EM | Dm | 1 | 1 | 1 | 1 | 2 | 1 | 4 |
| | *Nasonia vitripennis* | G | M | Nv | 1 | 1 | 1 | 1 | 2 | 1 | 4 |
| Arthropoda | *Tribolium castaneum* | G | M | Tc | 1 | 1 | 1 | 1 | 2 | 1 | 4 |
| | *Apis mellifera* | G | EM | Am | 1 | 1 | 1 | 1 | 2 | 1 | 4 |
| | *Lasioglossum albipes* | G | K | La | 1 | 1 | 1 | 1 | 2 | 1 | 4 |
| | *Acyrthosiphon pisum* | G | A | Ap | 1 | 1 | 1 | 1 | 2 | 1 | 15 |
| Platyhelmintes | *Schistosoma mansoni* | G | GD | Sch_man | 1 | 1 | 2 | 1 | 2 | 1 | 3 |
| | *Schmidtea mediterranea* | G | SG | Sch_med | 1 | 1 | | 1 | 2 | 1 | 4 |
| Rotifera | *Adineta vaga* | G | V | Adi_vag | 1 | 1 | 1 | 1 | 1 | 1 | 4 |
| Brachiopoda | *Lingula anatina* | G | L | Lin_ana | 1 | | 1 | 1 | 1 | 1 | 3 |
| Annelida | *Capitella telata* | G | EM | Ct | 1 | 1 | 1 | 1 | 1 | 1 | 3 |
| | *Helobdella robusta* | G | EM | Hel_rob | 1 | 1 | 1 | 1 | 1 | 1 | 4 |
| Cephalopoda | *Octopus bimaculoides* | G | M | Oct_bim | 1 | 1 | 1 | 1 | 1 | 1 | 4 |
| | *Aplysia californica* | G | B | Ac | 1 | 1 | 1 | 3 | 1 | 1 | 4 |
| | *Lottia gigantea* | G | M | Lg | 1 | 1 | 1 | 1 | 1 | 1 | 3 |
| | *Mytilus galloprovincialis* | T | Local | Mg | **1** | **1** | **1** | 1 | **1** | 1 | 3 |
| | *Mytilus edulis* | T | Local | Me | **1** | **1** | **1** | 1 | | 1 | 2 |
| | *Mytilus californianus* | T | Local | Mc | | | | | 1 | 1 | **2** |
| | *Mytilus trossulus* | T | Local | Mt | | | | | | 1 | |
| | *Anadara trapezia* | T | Local | At | | | | | 1 | | **1** |
| | *Tegillarca granosa* | T | Local | Tg | | | | | 1 | | |
| | *Bathymodiolus azoricus* | T | Local | Ba | | | | | 1 | | |
| | *Perna viridis* | T | Local | Pv | | | 1 | 1 | 1 | 1 | **2** |
| | *Ennucula tenuis* | T | Local | Et | | | | | 1 | 1 | **1** |
| | *Crassostrea corteziensis* | T | Local | Cc | | **1** | 1 | 1 | 1 | | **4** |
| | *Crassostrea gigas* | G | EM | Cg | 1 | 1 | 1 | 1 | 1 | 1 | 4 |

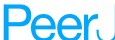

**Table 1** (*continued*)

| Phylum | Species | Sequence origin | Ref | Used ID | DROSHA | DGCR8 | XPO5 | RAN | DICER | TARBP2 | AGO or PIWI |
|---|---|---|---|---|---|---|---|---|---|---|---|
| | *Crassostrea hongkongensis* | T | Local | Ch | | 1 | | 1 | | | 3 |
| | *Crassostrea virginica* | T | Local | Cv | 1 | | | | | | 4 |
| | *Crassostrea angulata* | T | local | Ca | 1 | 1 | 1 | 1 | | | 2 |
| Mollusca | *Ostrea chilensis* | T | Local | Oc | | | | | 1 | | |
| | *Ostrea edulis* | T | Local | Oe | | | | | 1 | | 2 |
| | *Ostrea lurida* | T | local | Ol | | | | | 1 | | 1 |
| | *Ostreola stentina* | T | Local | Os | | | | | 1 | | |
| | *Saccostrea glomerata* | T | Local | Sg | | | | | 1 | | |
| | *Argopecten irradians* | T | Local | Ai | | | | | 1 | | 1 |
| | *Mizuhopecten yessoensis* | T | Local | My | 1 | | | 1 | 1 | 1 | 2 |
| | *Pecten maximus* | T | Local | Pm | | | | 1 | | | 2 |
| | *Pinctada fucata* | G | F | Pf | | | 1 | 1 | 1 | 1 | |
| | *Solemya velum* | T | Local | Sv | 1 | 1 | 1 | 1 | | 1 | 3 |
| | *Elliptio complanata* | T | Local | Ec | 1 | | 1 | 1 | | 1 | 1 |
| | *Pyganodon grandis* | T | Local | Pg | | | 1 | 1 | | | 2 |
| | *Uniomerus tetralasmus* | T | Local | Ut | | | 1 | 1 | | | 3 |
| | *Villosa lienosa* | T | Local | Vl | | | | 1 | | | 1 |
| | *Corbicula fluminea* | T | local | Cf | | | | | 1 | | 1 |
| | *Meretrix meretrix* | T | local | Mm | | | | | 1 | | 2 |
| | *Ruditapes decussatus* | T | local | Rd | | | | | 1 | | 1 |
| | *Ruditapes philippinarum* | T | local | Rp | | | | | 1 | | 1 |
| Echinodermata | *Strongylocentrotus purpuratus* | G | M | Sp | 1 | 1 | 1 | 1 | 1 | 1 | 3 |
| Hemichordata | *Saccoglossus kowalevskii* | G | M | Sk | 1 | 1 | 1 | 1 | 1 | 1 | 1 |
| | *Homo sapiens* | G | M | Hs | 1 | 1 | 1 | 1 | 1 | 1 | 8 |
| | *Ciona intestinalis* | G | M | Ci | 1 | 1 | 1 | 1 | 1 | 1 | 3 |
| Chordata | *Branchiostoma floridae* | G | M | Bf | 1 | 1 | 1 | 1 | 1 | 1 | 7 |
| | *Oncorhynchus mykiss* | G | O | Om | 1 | 1 | 1 | 1 | 1 | 1 | 5 |
| | *Danio rerio* | G | M | Dr | 1 | 1 | 1 | 1 | 1 | 1 | 5 |
| Streptophyta | *Arabidopsis thaliana* | G | P | At | No | No | 1 | 1 | 4 | No | 10 |
| | *Populus trichocarpa* | G | P | Pt | No | No | 1 | 1 | 5 | No | 11 |

**Notes.**

Abbreviations:: A, Aphidbase; B, broadinstitute.org/ftp/pub/assemblies/invertebrates/aplysia/; EM, Ensambl Metazoa v.29; F, Takeuchi et al. (2012) DNA Res. 19(2): 117–130;; G, Genome; GD, GeneDB; K, Kocher et al. (2013) Genome Biology 14 (12): R142; L, Lou et al. (2015) Nat Commun. 8; 6:8301; M, Metazome v3.0; O, Berthelot et al. (2014) Nat Commun. 22; 5: 3657; P, Phytozome 11; SG, SmedGD; T, Transcriptome; TSA, NCBI Transcriptome shotgun assembly; V, Genoscope.

**Table 2  Key proteins of the miRNA biogenesis with their structural domains.**

| Process step | Protein name | Domains |
|---|---|---|
| Microprocessor complex | DROSHA | 2× RIBO III + 1× DSRM |
| | DGCR8 | 1× WW + 2× DSRM |
| Moving to cytoplasm | XPO5 | 1× XPO1 |
| | RAN | 1× RAN |
| RISC loading complex | DICER | PDB + Helicase + DICER + PAZ + 2× RIBO + DSRM |
| | TARBP2 | 3× DSRM |
| Final miRNA maturation | AGO | DUF + PAZ + PIWI |
| | PIWI-like | PAZ + PIWI |

protein coding sequences (cds) were predicted with Transdecoder (*Grabherr et al., 2011*). Transcriptomic reads of 30 bivalve *spp.* (Cg plus other 29 species) were retrieved from the SRA archive and assembled as described above (details in File S1). The protein predictions of further 33 organisms were directly retrieved from public repositories or extracted from the corresponding genome releases. The NCBI transcriptome shotgun assembly (TSA) database was interrogated to retrieve hits for two additional cnidarians, *Porites australiensis* and *Anthopleura elegantissima* (Table 1).

## Protein domain searches

To investigate the presence of eight key proteins of miRNA biogenesis (DROSHA, DGCR8, XPO5, RAN, DICER, TARBP2, AGO and PIWI), we downloaded their predictive HMM from PFAM v.27 (listed in Table 2) and we scanned the sequence datasets with HMMer v3.1 (*Eddy, 2011*) applying a cut-off E-value of 0.01. To achieve a meaningful comparison of proteins from different organisms, we retained only hits presenting all diagnostic domains. Moreover, we identified several mussel transcripts related to protein interactions occurring in the miRNA biogenesis. To identify such proteins, we retrieved from PFAM the diagnostic domains of human homologs (listed in Table 3) and we scanned their presence in the Mg transcriptome as described above. Protein domain organization was reconstructed using SMART (*Letunic, Doerks & Bork, 2012*).

## Gene structure analysis

We used the transcript sequences of DROSHA, DGCR8, XPO5, DICER and TARBP2 as blast queries against all Mg genomic contigs (*blastn*) in order to recover the related gene structures. Positive hits having $e$-value lower than $10^{-20}$ were extracted and assembled on the corresponding transcript, used as backbone. RNA-seq read mappings with adapted parameters (CLC Genomic Workbench *large gap mapping tool*, with similarity and length fraction set at 0.9) allowed us to ascertain the correct gene assembly. Homolog gene structures were retrieved by interrogating genomic browsers, like Metazome v.3 (for *C. intestinalis, B. floridae, D. rerio, S. kowalevskii, S. purpuratus, N. vectensis, T. castaneum, L. gigantea, O. bimaculoides, C. elegans* and *H. sapiens*) and Ensembl Metazoa v.29 (for *C. gigas, C. quinquefasciatus, D. melanogaster, N. vitripennis, A. mellifera, A. queenslandica,*

Rosani et al. (2016), *PeerJ*, DOI 10.7717/peerj.1763

**Table 3  miRNA biogenesis proteins of *Mytilus galloprovincialis*.** Protein name, GenBank ID, transcript (bp) and protein length (aa), identified domains and annotation (first hit, *e*-value and percentage of similarity) are reported.

| | Protein name | GenBank ID | Transcript length (bp) | Protein length (aa) | Identified domain(s) | Annotation (first hit) | *E*-value (e^) | Similarity (%) |
|---|---|---|---|---|---|---|---|---|
| Key miRNA biogenesis proteins | MgDROSHA | KT447251 | 4,384 | 1,377 | 2× RIBO III + 1× DSRM | Ribonuclease 3-like (*Crassostrea gigas*) | 0 | 67 |
| | MgDGCR8 | KT447252 | 2,483 | 728 | 1× WW + 2× DSRM | Microprocessor complex subunit DGCR8-like (*Crassostrea gigas*) | 0 | 50 |
| | MgXPO5 | KT447259 | 3,875 | 1,201 | XPO1 | Exportin-5-like (*Crassostrea gigas*) | 0 | 55 |
| | MgRAN | KT447254 | 1,113 | 214 | RAN | GTP-binding nuclear protein Ran (*Crassostrea gigas*) | −143 | 93 |
| | MgDICER | KT447258 | 6,013 | 1,850 | PDB + Helicase + DICER + PAZ + 2× RIBO + DSRM | Endoribonuclease Dicer-like (*Crassostrea gigas*) | 0 | 58 |
| | MgTARBP2 | KT447253 | 7,583 | 321 | 3× DSRM | Probable RISC-loading complex subunit (*Crassostrea gigas*) | −143 | 69 |
| | MgAGO | KT447257 | 3,337 | 892 | DUF + PAZ + PIWI | Protein argonaute-2-like (*Crassostrea gigas*) | 0 | 84 |
| | MgPIWIa | KT447255 | 2,686 | 867 | PAZ + PIWI | Piwi-like protein 1 (*Crassostrea gigas*) | 0 | 75 |
| | MgPIWIb | KT447256 | 3,603 | 948 | PAZ + PIWI | Piwi-like protein 2 (Hydra vulgaris) | 0 | 59 |
| | MgGW182 | KT447250 | 3,825 | 1,274 | UBA + RRM | Trinucleotide repeat-containing gene 6C protein-like (*Crassostrea gigas*) | 0 | 45 |
| | MgCNOT1 | KT694355 | 5,373 | 1,791 | DUF3819 + NOT1 | CCR4-NOT transcription complex subunit 1-like (*Crassostrea gigas*) | 0 | 69 |
| | MgCNOT2 | KT694357 | 864 | 288 | NOT2_3_5 | CCR4-NOT transcription complex subunit 2 (Pinctada fucata) | −156 | 82 |
| | MgCNOT3 | KT694358 | 2,142 | 714 | NOT3 + NOT2_3_5 | CCR4-NOT transcription complex subunit 3-like (*Crassostrea gigas*) | 0 | 97 |
| | MgCNOT6 | KT694359 | 2,592 | 864 | Exo_endo_phos | Uncharacterized protein LOC105348954 isoform X1 (*Crassostrea gigas*) | 0 | 71 |
| | MgCNOT7 | KT694360 | 897 | 299 | CAF1 | CCR4-NOT transcription complex subunit 7-like (*Crassostrea gigas*) | 0 | 84 |
| | MgCNOT9 | KT694361 | 927 | 309 | RCD1 | Cell differentiation protein RCD1 homolog (*Crassostrea gigas*) | 0 | 93 |
| | MgCNOT10 | KT694356 | 2,133 | 711 | TPR_1 | CCR4-NOT transcription complex subunit 10-like (*Crassostrea gigas*) | 0 | 71 |
| | MgDDX5 | KT694371 | 1,740 | 538 | DEAD + Helic | ATP-dependent RNA helicase DDX5 (*Crassostrea gigas*) | 0 | 75 |
| | MgDDX6 | KT694372 | 1,332 | 443 | DEAD + Helic | ATP-dependent RNA helicase me31b (*Crassostrea gigas*) | 0 | 88 |

Rosani et al. (2016), *PeerJ*, DOI 10.7717/peerj.1763

**Table 3** (*continued*)

| | Protein name | GenBank ID | Transcript length (bp) | Protein length (aa) | Identified domain(s) | Annotation (first hit) | *E*-value (e^) | Similarity (%) |
|---|---|---|---|---|---|---|---|---|
| Other interacting proteins | MgDDX20 | KT694373 | 1,836 | 612 | DEAD + Helic | ATP-dependent RNA helicase DDX20 (*Crassostrea gigas*) | 0 | 77 |
| | MgDDX42 | KT694374 | 2,196 | 731 | DEAD + Helic | ATP-dependent RNA helicase DDX42 (*Crassostrea gigas*) | 0 | 72 |
| | MgPABP | KT694365 | 1,881 | 627 | 4× RRM + PABP | polyadenylate-binding protein 4 (Hydra vulgaris) | 0 | 74 |
| | MgeIF4G | KT694364 | 5,019 | 1,672 | MIF4G + MA3 + W2 | eukaryotic translation initiation factor 4 gamma (*Crassostrea gigas*) | 0 | 57 |
| | MgPAN2 | KT694367 | 3,606 | 1,202 | UCH_ 1 + RNase_T | PAB-dependent poly(A)-specific ribonuclease sub-unit PAN2 (*Lingula anatina*) | 0 | 72 |
| | MgPAN3 | KT694368 | 2,334 | 778 | None | PAB-dependent poly(A)-specific ribonuclease sub-unit PAN3 (*Lingula anatina*) | 0 | 67 |
| | MgDCP1 | KT694362 | 1,611 | 536 | DCP1 | mRNA-decapping enzyme 1A-like (*Crassostrea gigas*) | −126 | 73 |
| | MgDCP2 | KT694363 | 1,313 | 385 | DCP2 + NUDIX | m7GpppN-mRNA hydrolase (*Lingula anatina*) | −117 | 67 |
| | MgPRMT5 | KT694369 | 1,893 | 631 | PRMT5 | protein arginine N-methyltransferase 5-like (*Crassostrea gigas*) | 0 | 72 |
| | MgTudor-11 | KT694370 | 2,682 | 894 | 4× SNc + TUDOR | Hypothetical protein mRNA (*Lottia gigantea*) | 0 | 73 |
| | MgMaelstrom | KT694366 | 1,321 | 404 | HMG + MAEL | Protein maelstrom (*Crassostrea gigas*) | −155 | 62 |

*P. bachei, M. leidyi, T. adhaerens, N. vectensis, D. pulex, S. mansoni, S. mediterranea, A. vaga, L. anatine, H. robusta* and *C. telata*) or by local *blastn* against the downloaded genomes (A. *pisum* and *L. albipes*).

### Phylogenetic analysis

The inferred protein sequences were aligned using MUSCLE, release 2014-05-29 (*Edgar, 2004*). Subsequently, the fasta alignments were analyzed using Gblocks v.0.91 (*Castresana, 2000*) to extract conserved positions (positions common to 51% of the locally aligned sequences). Trees were built using neighbor joining or maximus likelihood clustering methods with 1,000 bootstrap replicates. Bayesian phylogenies were reconstructed using MrBayes v.3.2.5 (*Ronquist et al., 2012*), with GTR substitution evolutionary model with gamma-distributed rate variation across sites, evaluating the convergence after 1,000,000 runs (0.5 was considered as cut-off value). Trees were visualized and edited with FigTree v1.4.2 (http://tree.bio.ed.ac.uk/software/figtree/).

### Digital expression analysis

To analyze the expression of the selected genes in Cg and Mg RNA datasets, we retrieved all available RNA-seq samples from the NCBI SRA archive. For Cg, we analyzed 123 Illumina RNA-seq samples related to adult tissues or developmental stages. For Mg, we analyzed 13 RNA samples from gills (1), digestive gland (6), haemocytes (2), mantle (2) and muscle (2). Overall, we included in the expression analysis 2,271 and 453M reads for Cg and Mg, respectively (File S2). The trimmed reads were mapped to Cg and Mg genes using the CLC Genomics Workbench v.8.0 (Qiagen, Hilden, Germany) mapping tool, with length and similarity fractions set at 0.75 and 0.95, respectively, and mismatch/insertion/deletion penalties at 3/3/3. The number of uniquely mapped reads of each dataset were counted and used to calculate digital expression values as TPM (Transcripts Per Kilobase Million mapped reads) as described by (*Wagner, Kin & Lynch, 2013*), considering 3 TPMs as lower detection limit.

## RESULTS

### Mussel transcripts related to the miRNA biogenesis

We identified *Mytilus galloprovincialis* transcripts involved in the miRNA biogenesis by systematic searches of diagnostic domains (Table 2) in a transcriptome assembly produced from 453 million Illumina reads. Thus, we recovered nine transcripts coding for DROSHA, DGCR8, XPO5, RAN, DICER, TARBP2 and for three Argonaute genes (one Ago and two Piwi-like proteins, Table 3). We also identified 21 mussel proteins expected to play a role in the miRNA maturation or involved in RNAi processes (File S3).

Figure 1 relates the general process of eukaryotic miRNA biogenesis to the mussel proteins identified in this work. MgDROSHA and MgDGCR8 are expected to start the maturation of pri-miRNAs produced by RNA polymerase II. MgDROSHA codes for a 1,377 aa length protein containing all the canonical domains (2 RIBOc domains in positions 959–1,093 and 1,139–1,271 and one DSRM domain in position 1,278–1,351) whereas MgDGCR8 is a 728 aa length protein having one WW domain in position 229–258,

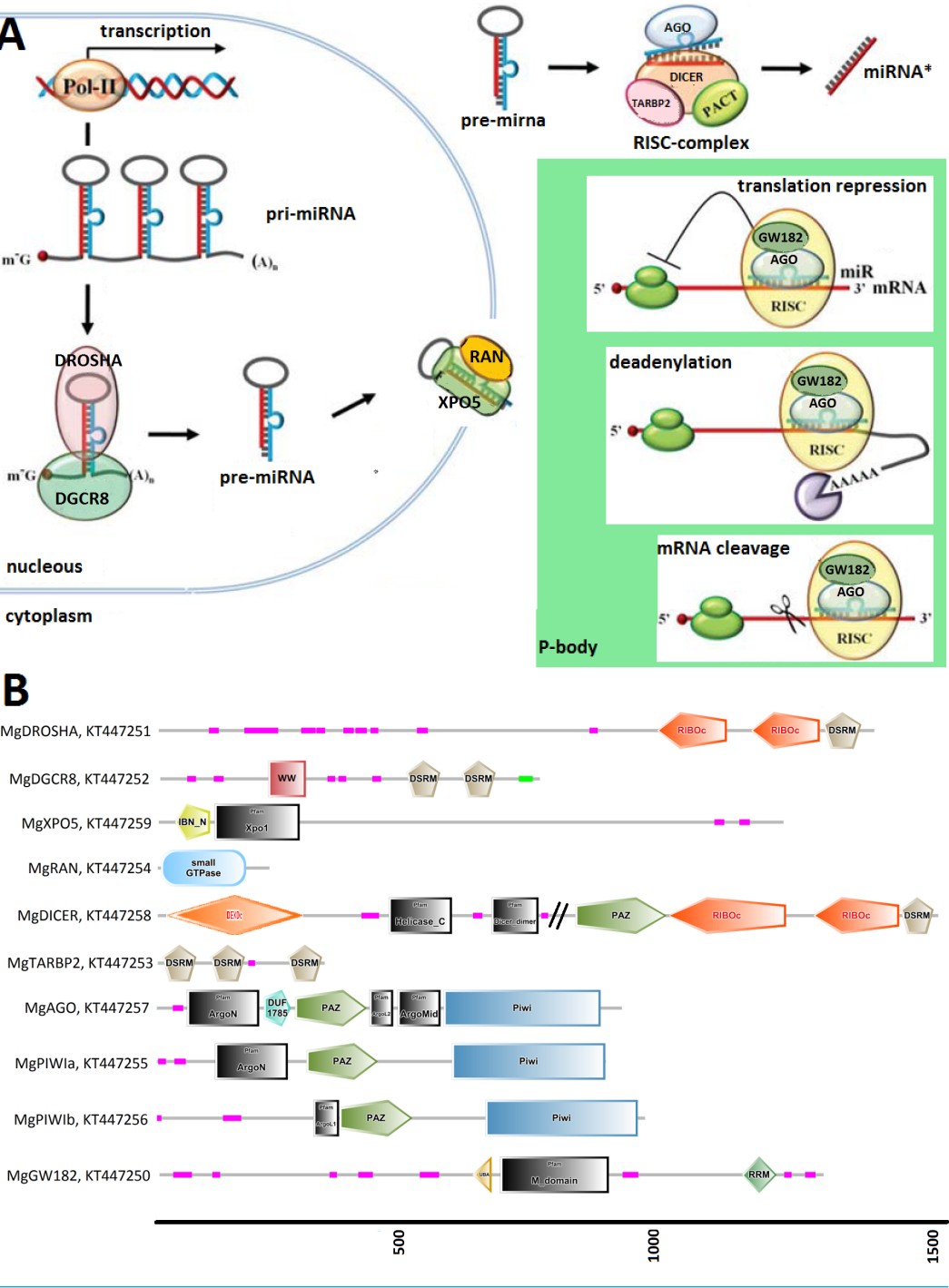

**Figure 1** (A) **Graphical reconstruction of mussel miRNA biogenesis process.** (Modified from *Kapinas & Delany, 2011*). (B) Conserved domains of the mussel miRNA complements.

necessary for the interaction with DROSHA, and two DSRM domains (positions 472–536 and 578–642) necessary for pri-miRNA binding. MgXPO5 is expected to cooperate with MgRAN in the pre-miRNA cytoplasmic translocation. MgRAN encodes a 214 aa protein whereas MgXPO5 has a length of 1,201 aa and includes two 5' conserved domains (IBN_N and Xpo1) and one conserved region necessary for the interaction with interleukin enhancer-binding factor 3 (position 525–562). In mussels, the RISC complex uploading pre-miRNAs is defined by the endoribonuclease MgDICER (1,850 aa) and MgTARBP2 (321 aa). Like in *Lophotrocozoa*, mussel DICER is encoded by a unique gene and contains the seven canonical domains, namely two helicase domains, one DICER-dimer domain, one PAZ, two RIBOc and a final DSRM domain. MgTARBP2 displays three DSRM domains in positions 9–73, 101–166 and 249–314. Moreover, *M. galloprovincialis* possess three argonaute proteins ranging from 861 to 941 aa in length and representative of one AGO (DUF1785, PAZ and PIWI domains) and two PIWI-like proteins (PAZ and PIWI domains). We considered the above mentioned gene products as the key complement of the miRNA biogenesis.

Among the possible interacting proteins, we identified MgGW182, a transcript encoding a protein shorter than the human counterparts but holding all the features considered significant for its interaction with AGOs and the CCR4-NOT complex. In fact, MgGW182 possesses 19 N-terminal GW stretches, followed by one UBA domain, a Q-rich region (M domain) and a C-terminal RNA recognition motif (RRM domain). Moreover, we recognized a C-terminal conserved site known as PAM2 (*Kozlov et al., 2010*), expected to interact with the poly(A) binding protein 1 (MgPABPC1) through the MLLE motif and inhibit the mRNA translation by interfering with the mRNA circularization process (*Piao et al., 2010*; *Van Kouwenhove, Kedde & Agami, 2011*). In the mussel transcriptome, we also found putative homologs for a number of CNOT complex proteins (CNOT1, 2, 3, 6, 7, 9, and 10), for the eukaryotic translation initiation factor 4 gamma, 1 eIF4G, PAB-dependent poly(A)-specific ribonuclease subunits PAN2, PAN3, the decapping complex proteins DCP1 and DCP2, and several RNA helicases demonstrated to be crucial in the miRNA maturation (DDX5) and RNAi (DDX5- 6- 20 and 42). Finally, we recognized the putative mussel homologs of protein arginine methyltransferase 5 (MgPRMT5), tudor domain containing protein (MgTDRD-11) and maelstrom spermatogenic transposon silencer (MgMAEL).

## Mussel genes related to the miRNA biogenesis

Taking advantage of mussel WGS data (*Nguyen, Hayes & Ingram, 2014*) we investigated the organization of the main genes involved in the mussel miRNA biogenesis. Fragmentation of the genomic mussel assembly (2.3 million contigs; 700 bp on average) and considerable dimension of the analyzed genes (9.6–17.6 kbp gene size in the case of Cg) prevented the recovery of the full gene sequences. Nevertheless, we can describe the complete gene structures of DROSHA, DGCR8, EXP5, DICER and TARBP2 (i.e., five of eight searched sequences) whose length varies between 7.5 and 27 kbp, confirmed by the back-mapping of 115,377 Illumina paired reads (Fig. 2, File S4). Moreover, these mussel genes showed
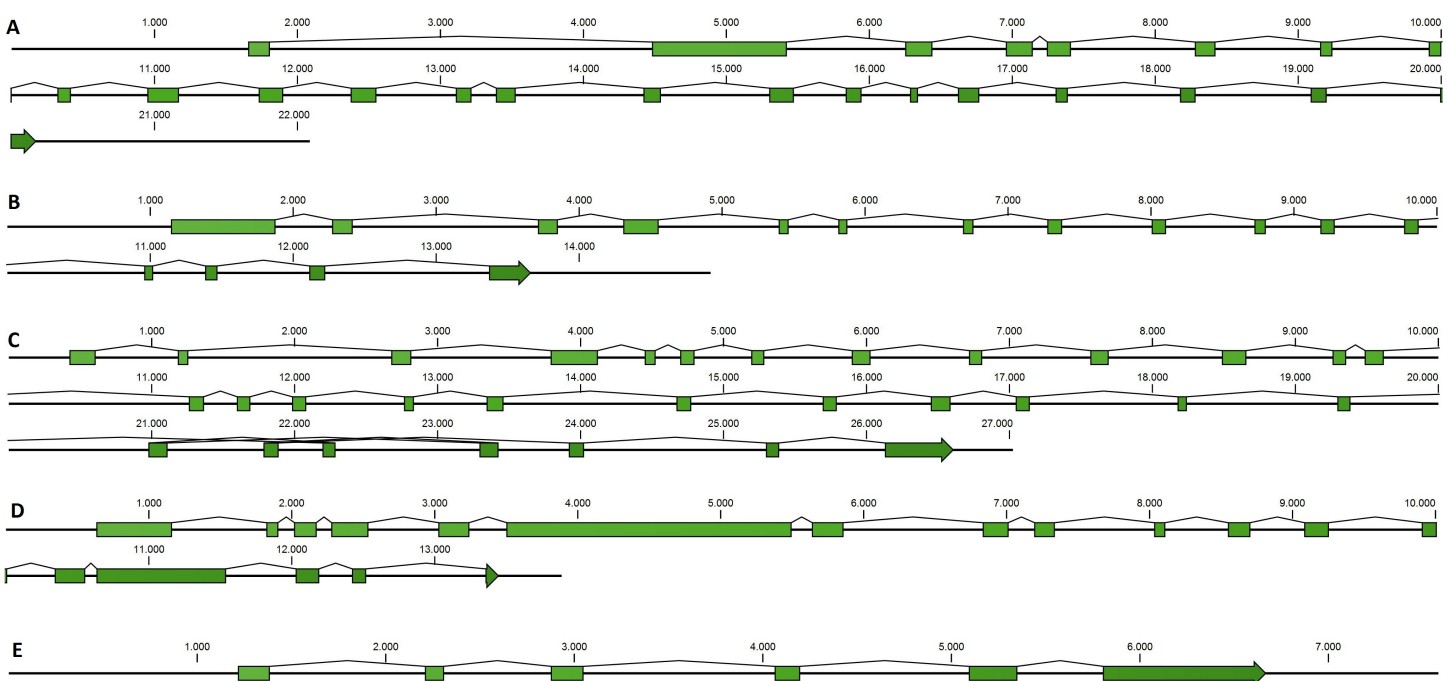

**Figure 2** **Mussel gene structures of DROSHA (A), DGCR8 (B), EXP5 (C), DICER (D) and TARBP2 (E).** Green boxes represent exons, length is reported as base pair scale.

a remarkable conservation in terms of exon number when compared with a selection of homolog genes from deuterostome and protostome organisms (Table 4).

## Transcripts related to the miRNA biogenesis in bivalve *spp*

To identify the miRNA biogenesis complements in marine mollusks, we used homologous genes retrieved from the genomes of *C. gigas, L. gigantea* and *A. californica*. Since the *C. gigas* genome includes annotations only for the cds regions, we exploited full-length transcripts obtained from a locally assembled oyster transcriptome to expand the genome annotations in this species. In particular, we updated the annotation of CgDGCR8 and CgDICER and we added new annotations for CgPIWI-1 (CGI_10008757: genomic contig JH815696, position 184178–187825) and CgTARBP2 (JH818440, 414703–419857).

Since many marine bivalve *spp.* do not have at present a sequenced genome, we used publicly available RNA-seq data to build 29 specie-specific transcriptome assemblies and retrieve the homologous sequences of interest. After domain searching, we carefully considered the high number of positive hits to retain only proteins including all the expected protein features. Thus, we retrieved 132 complete hits from marine mollusks: 10 DROSHAs, 9 DGCR8s, 14 XPO5s, 34 RANs, 7 DICERs, 13 TARBP2s and 45 Argonaute-like proteins, the latter classified in 13 AGO and 32 PIWI proteins by phylogenetic analysis (Table 1, File S5).

## Phylogenetic analysis of the miRNA biogenesis proteins

The inferred sequences of single miRNA biogenesis proteins were aligned together with those retrieved from 34 sequenced genomes. Here, we report the phylogenetic analysis of

**Table 4  Number of exons of five key miRNA biogenesis genes.** Metazome 3.0 and Ensembl Metazoa v.29 genome browsers were interrogated with the previously analyzed hits for each organism. La and Ap genomes were downloaded and analyzed locally. Mg gene structures were retrieved as described in Methods. In green are reported Protostomia; in orange Deuterostomia.

| Species | *Homo sapiens* | *Ciona intestinalis* | *Branchiostoma floridae* | *Danio rerio* | *Saccoglossus kowalevskii* | *Strongylocentrotus purpuratus* | *Nematostella vectensis* | *Amphimedon queenslandica* | *Caenorhabditis elegans* | *Capitella telata* | *Culex quinquefasciatus* | *Drosophila melanogaster* | *Nasonia vitripennis* | *Tribolium castaneum* | *Apis mellifera* | *Lasioglossus albipes* | *Acyrthosiphon pisum* | *Lottia gigantea* | *Crassostrea gigas* | *Mytilus galloprovincialis* |
|---|---|---|---|---|---|---|---|---|---|---|---|---|---|---|---|---|---|---|---|---|
| ID | Hs | Ci | Bf | Dr | Sk | Sp | Nvec | Aq | Ce | Ct | Cq | Dm | Nv | Tc | Am | La | Ap | Lg | Cg | Mg |
| DROSHA | 27 | 24 | 29 | 17 | 24 | 20 | 13 | 14 | 6 | 28 | 3 | 3 | 11 | 9 | 13 | 23 | 1 | 23 | 30 | 23 |
| DGCR8 | 14 | 10 | 15 | 10 | 15 | 13 | 7 | No | 11 | 18 | 4 | 5 | 11 | 6 | 6 | 6 | 2 | 11 | 18 | 16 |
| XPO5 | 32 | 4 | 8 | 21 | 29 | 31 | 2 | 28 | No | 30 | 9 | 2 | 11 | 10 | 9 | 10 | 1 | 32 | 34 | 31 |
| DICER | 27 | 23 | 26 | 17 | 14 | 19 | 12 | 10 | 26 | 13 | 7 | 8 | 5 | 9 | 29 | 33 | 19 | 16 | 19 | 18 |
| TARBP2 | 9 | 1 | 6 | 9 | 6 | 2 | No | No | 11 | 8 | 4 | 5 | 7 | 7 | 7 | 6 | 7 | 6 | 7 | 6 |

the five proteins centrally involved in the miRNA biogenesis, namely DROSHA, DGCR8, DICER, TARBP2 and AGOs (File S5 includes all protein sequences). We back-traced the presence of a canonical DROSHA up to Cnidaria, although we found only incomplete hits in Porifera and Placozoa and the genomes of *Ctenophora spp.* lack of both DROSHA and DGCR8, as reported by other authors (*Maxwell et al., 2012*). The DROSHA sequences from Cnidaria's appeared as general outgroup whereas those of Chordata clustered as outgroup of the other protostomes. DROSHAs from Mollusca and Arthropoda clustered consistently with the different *taxa* whereas those from Platyhelmintes, Rotifera, Brachiopoda and Annelida grouped together, with DROSHA from *Caenorhabditis elegans* (Nematoda) being the most far-related (Fig. 3A). Contrary to DROSHA, we identified a complete DGCR8 also in the *Porifera Amphimedon queenslandica*, suggesting that also DROSHA should be present in this taxa. Following phylogenetic analysis, we highlighted Cnidaria and Porifera proteins as outgroup, with mollusks (and Annelida) clustering with Arthropoda and more distantly *Platyhelmintes* and *Rotifera* hits. The *Chordata* sequences clustered as a separate group (Fig. 3B).

The finding of putative DICER sequences in Ctenophora *spp.* supports the presence of this gene through the whole Opisthokonta evolution (*Maxwell et al., 2012*). Also plants possess DICER homologues which occur in different copy number among taxa: two genes in Porifera, Placozoa, Cnidaria, Platyhelminthes and Arthropoda (with the exception of *D. pulex* that possess three genes); four genes in plants like *A. thalian*a and *P. trichocarpa* and one gene in Ctenophora, Rotifera, Cephalopoda Mollusca and Chordata. Moreover, the presence of DICER was reported in some Protozoa and fungi (*Mukherjee, Campos & Kolaczkowski, 2013*). Phylogenetic analyses, separate insect DICER-2,plant DICERs from DICER-1. DICER-1 clade shows a consistent clustering of Arthropoda, Mollusca

Rosani et al. (2016), *PeerJ*, DOI 10.7717/peerj.1763

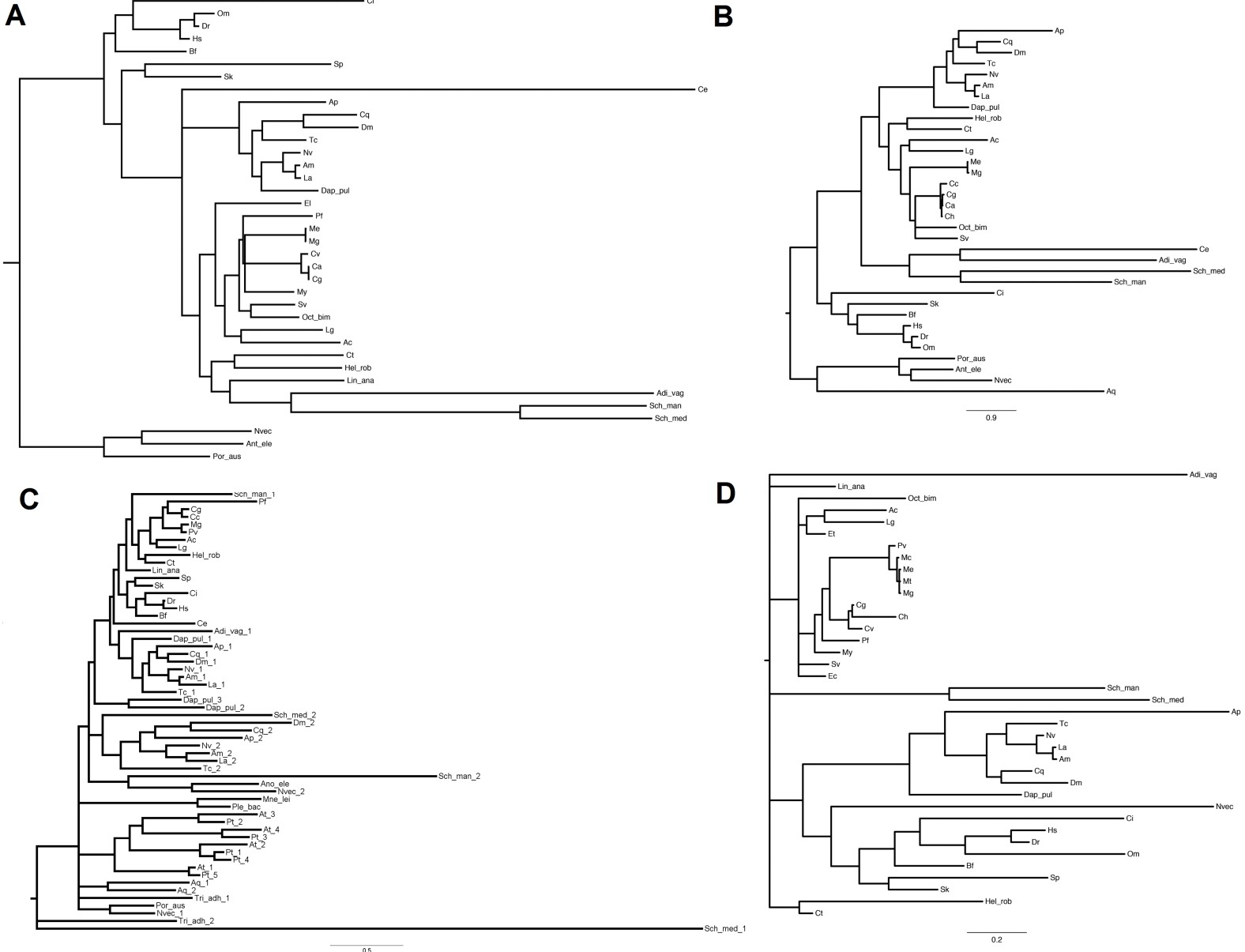

**Figure 3** **Phylogenetic relationships of four miRNA biogenesis proteins.** (A) DROSHA, (B) DGCR8, (C) DICER and (D) TARBP2. Inferred protein sequences were aligned using MUSCLE, conserved positions were extracted using Gblocks and subjected to MrBayes analysis.

and Chordata hits, whereas some branches of basal metazoans and Platyhelminthes are not well resolved (Fig. 3C). Likewise, the phylogenetic tree regarding TARBP2 displays a clear cut-off between the proteins of mollusks, chordates and arthropods (Fig. 3D). We back-traced the miRNA cytoplasm export complex composed by RAN and XPO5 in all analyzed metazoans. Both RAN and XPO5 represent widely expressed sequences since we found them also in transcriptome assemblies, although with suboptimal sequence coverage.

Several AGO and Piwi proteins can be present in individual organisms and, in fact, we identified a total of 235 proteins. Whereas humans possess eight proteins, we found four proteins in the majority of the analyzed insect *spp.* (with the exception of 15 proteins in *A. pisum*) and three or four different proteins in bivalve *spp*. Also, basal Metazoa possess Argonaute-like sequences: four in the genomes of Ctenophora and Cnidaria *spp.*, one in the Placozoa *T. adhaerens* and two in *A. queenslandica*. The case of *C. elegans* is remarkable since it holds several Argonaute gene families and at least 24 proteins (*Hoogstrate et al., 2014*). In agreement with other phylogenetic studies (*Swarts et al., 2014*), the Argonaute proteins from plants and the majority of those from *C. elegans* formed distinct clades and, moreover, a clear separation was evident between AGO and PIWI proteins. Bivalve protein sequences clustered always separately forming one cluster for AGO-like hits and two clusters for PIWI-like proteins (Fig. 4).

## Digital expression analysis of mussel and oyster miRNA biogenesis genes

We used the 13 Mg and 124 Cg RNA-seq samples to evaluate the expression levels of miRNA biogenesis genes in different tissues and conditions. Based on total mapped reads, we computed TPM values and we used elongation factor 1 $\alpha$ (El1$\alpha$) as normalizer housekeeping gene to compare the expression level of the different genes in each sample.

For Mg, the sequence analysis indicated a scarce basal expression of the genes mentioned above in five adult tissues: gill, digestive gland, haemolymph, muscle and mantle (below 2% of El1$\alpha$, except for DDX5, RAN and CNOT9). Mantle and muscle appeared the most responsive tissues whereas haemolymph was the least responsive one. In particular, the genes that we considered as the core components of miRNA biogenesis were expressed at levels below 0.5% of El1$\alpha$ (File S6).

For Cg, we analyzed a considerable number of RNA-seq libraries representative of adult tissues (85) and developmental stages (39) (File S6). In adult oysters we observed low basal expression, as detected in the mussel samples. In fact, none of the experimental conditions reported for the analyzed RNA-seq samples influenced substantially the expression of the core miRNA pathway genes (expression levels below 2% of El1$\alpha$), with the exception of the high levels of CgPIWI-1 levels in male and female gonads (around 3.5%, Fig. 5). Conversely, most of the miRNA biogenesis genes were expressed at remarkable levels during the early stages of the oyster development: mainly from two cells to the rotary movement and, for some genes, also in the next developmental stages until *D-shaped* larvae, with no detectable signals afterward in spat and juveniles. Hence, these genes are particularly active in the early development, in particular one AGO (CGI_10020511) and two PIWI transcripts from the egg to trocophora (Fig. 5). In the same developmental stages we also noticed a

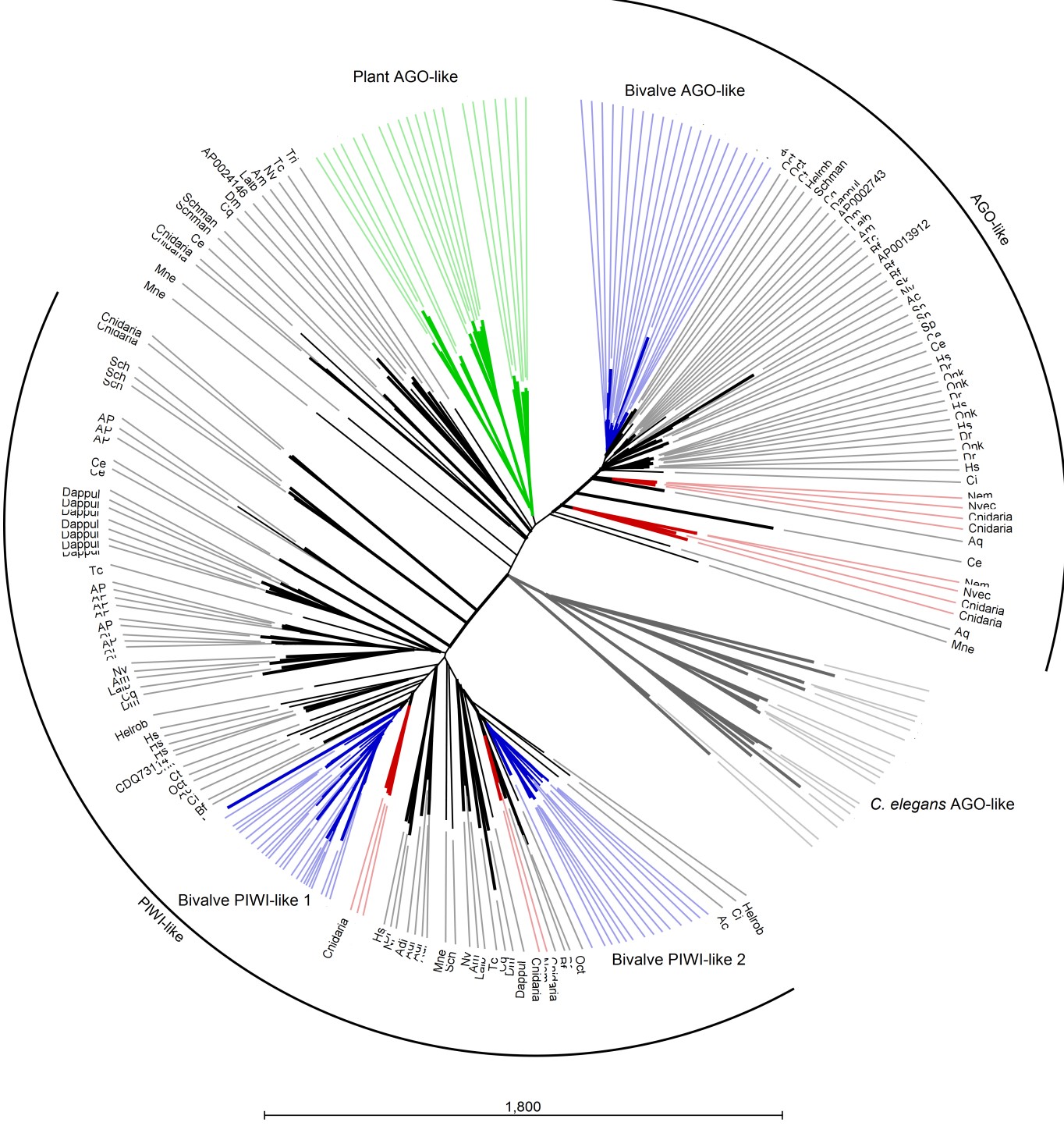

**Figure 4** **Phylogenetic relationships of Argonaute-like proteins.** Proteins were aligned using MUSCLE and tree was generated using Neighbor Joining algorithm with 1,000 bootstrap replicates. Plant proteins are highlighted in green, whereas *C. elegans* hits are reported in grey. Blue lines represent mollusk hits, red lines represent hits from basal metazoans.

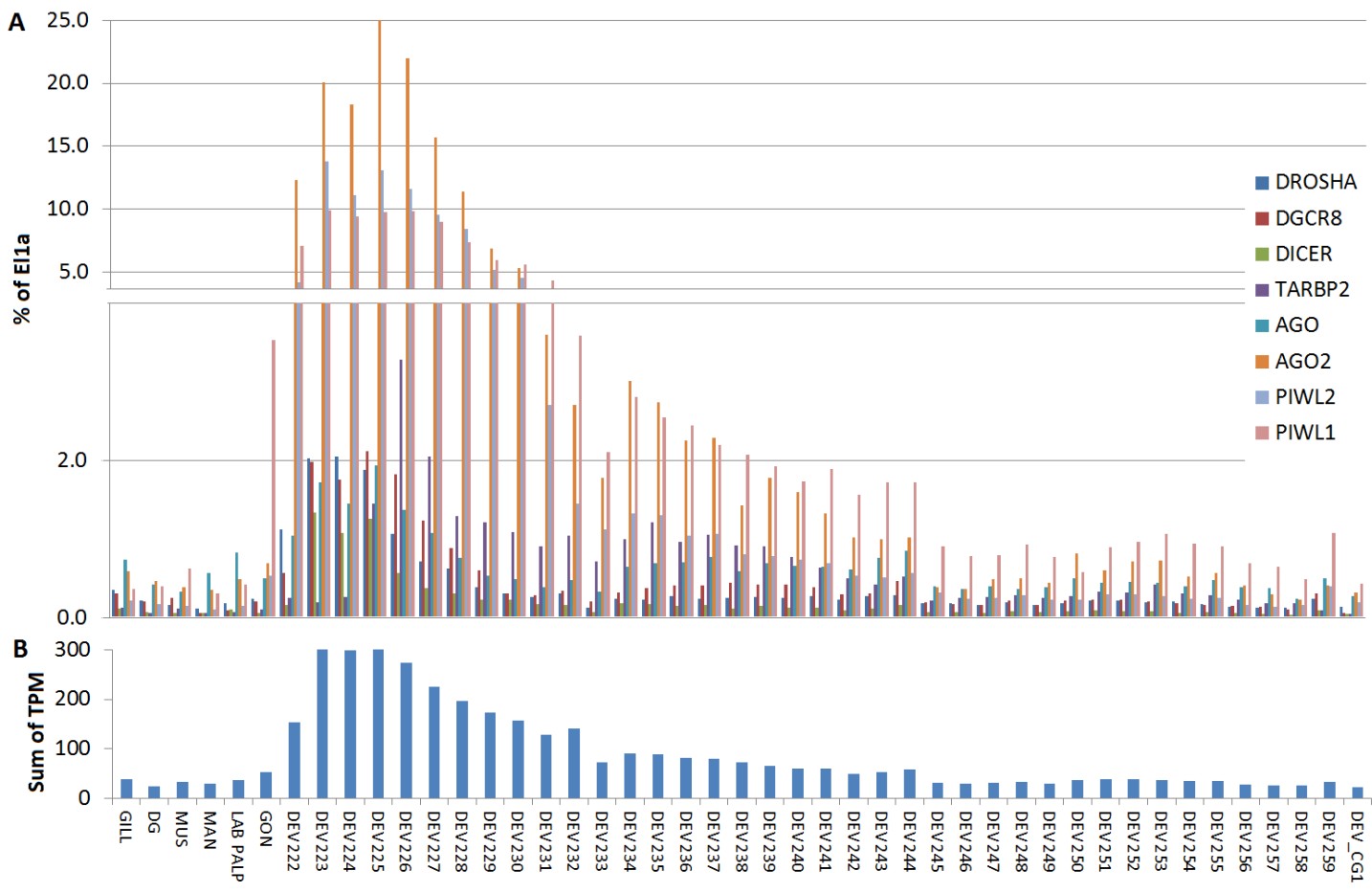

**Figure 5  Digital expression analysis in oyster.** The expression of the 8 miRNA biogenesis genes were computed in tissue-specific RNA libraries and in RNA libraries from different developmental phases. (A) Expression values represented as percentage of El1α. (B) Cumulative TPM expression values of the 8 genes in the same samples.

remarkable expression of the key miRNA genes, with the co-expression of DROSHA and DGCR8 evident in all the analyzed samples.

## DISCUSSION

Small RNAs are important regulators of the gene expression, as recognized in various model and non-model organisms (*Kim et al., 2014a*; *Kim et al., 2014b*; *Martini et al., 2014*; *Hussain & Asgari, 2014*; *Sahoo et al., 2014*; *Britton et al., 2014*; *Poole et al., 2014*; *Solofoharivelo et al., 2014*), including some bivalves (*Jiao et al., 2014*; *Zhou et al., 2014*). In addition to the identification of miRNAs, a general comprehension of the miRNA biogenesis in itself is also significant (*Grimson et al., 2008*; *Wu et al., 2011*; *Moran et al., 2013*). However, the main genes involved in miRNA formation in bivalves have not been described and characterized so far. In this study, we have provided an overview on the miRNA biogenesis complements in bivalves *spp.*, with particular attention to *M. galloprovincialis* and *C. gigas*. To the best of our knowledge, we report for the first

time the presence of a complete miRNA biogenesis pathway in *M. galloprovincialis,* the full-length transcript sequences of DICER, DGCR8, XPO5, RAN, DROSHA, TARBP2, three Argonaute genes and the identification of many other components that are candidate miRNA complement-interacting proteins such as MgGW182. By using local transcriptome assemblies, we identified these genes also in many other marine bivalves. The general low expression levels of these transcripts in the adult tissues of both *M. galloprovinciali*s and *C. gigas,* and the considerable gene size, have probably prevented a previous identification of full-length sequences in not-well-covered bivalve transcriptomes. In fact, we obtained complete transcript sequences only from sequenced genomes or highly-covered transcriptomes whereas in other transcriptome assemblies we retrieved only few complete sequences. Overall, we have analyzed 523 miRNA complement sequences, 145 of them belonging to marine mollusks and displaying a consistent sequence clustering (*Ostreoida* and *Mytiloida* proteins generated two distinct clades, located always as sister group of arthropods).

However, the copy number of Argonaute genes somewhat differs among bivalves, as *C. gigas* and *A. californica* genomes coding for four proteins (2 AGO and 2 PIWI proteins) whereas *M. galloprovincialis* and *L. gigantea* possess three proteins (1 AGO and 2 PIWIs). We also highlighted the over-expression of the miRNA biogenesis genes during the first phases of the oyster development. A genome protection mechanism based on piRNA expression during early developmental stages is well known in mammals (*Malone & Hannon, 2009*; *Kim et al., 2014a*; *Kim et al., 2014b*) but such mechanism has not been reported in bivalves and additional investigations are necessary.

Finally, the identification of several mussel proteins either necessary or cooperative in the miRNA biogenesis, supports the existence of a complete and functional miRNA pathway in mussels and, probably, in other bivalves. Up to now, protein–protein or protein-RNA interaction data are not available for bivalve *spp*. and these topics may represent a direction of work in the future. Meanwhile, the expression analyses of miRNA biogenesis genes coupled with the identification of the miRNAs expressed in naturally infected and laboratory-treated bivalves could provide both validation and new insights on these interesting processes.

### Funding

This work is supported by PRIN 2010-11 (20109XZEPR) and FP7-KBBE-2010-4-266157 (BIVALIFE). The funders had no role in study design, data collection and analysis, decision to publish, or preparation of the manuscript.

### Grant Disclosures

The following grant information was disclosed by the authors:
PRIN2010–11: 20109XZEPR.
BIVALIFE: FP7-KBBE-2010-4-266157.

## Competing Interests

The authors declare there are no competing interests.

## Author Contributions

- Umberto Rosani conceived and designed the experiments, performed the experiments, analyzed the data, wrote the paper, prepared figures and/or tables.
- Alberto Pallavicini conceived and designed the experiments, wrote the paper, reviewed drafts of the paper.
- Paola Venier conceived and designed the experiments, contributed reagents/materials/-analysis tools, wrote the paper, reviewed drafts of the paper.

## DNA Deposition

The following information was supplied regarding the deposition of DNA sequences:

Genbank: KT447251, KT447252, KT447259, KT447254, KT447258, KT447253, KT447257, KT447255, KT447256, KT447250, KT694355, KT694356, KT694357, KT694358, KT694359, KT694360, KT694361, KT694362, KT694363, KT694364, KT694365, KT694366, KT694367, KT694368, KT694369, KT694370, KT694371, KT694372, KT694373, KT694374.

## Data Availability

The research in this article did not generate any raw data.

## Supplemental Information

Supplemental information for this article can be found online at http://dx.doi.org/10.7717/peerj.1763#supplemental-information.

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
