# Peer review of "The miRNA biogenesis in marine bivalves"

_PeerJ, doi:10.7717/peerj.1763_

## Round 0.1 · original submission · Major Revisions

Overall this was an interesting manuscript that I enjoyed reading. The main sections that require attention are the methods and discussion. In particular, the methods and experimental design section is the weakest part of the manuscript. More information needs to be added to the methods section and the choice of parameters for analysis needs to be justified. Please check all three reviews for this information. The discussion is also overly speculative in many places and needs to be toned down and/or caveats added (see reviewers comments). All three reviewers have done a terrific job of providing comments to improve your manuscript, so please take the time to address all of their comments.

Reviewer 1 ·

Basic reporting

No Comments

Experimental design

No Comments

Validity of the findings

No Comments

Additional comments

In this manuscript, the authors proposed a glance on the phylogeny and expression of genes involved in the miRNA biogenesis in M. galloprovincialis and C. gigas. However, the methods used are not accurately described for repeat and there is much room for additional studies or deep discussion to bring the present study to be considered for publication in Peer J. Some major concerns should be depicted clearly.
Methods:
1. In this article, the authors compared genes obtained from 31 bivalve transcriptomes and 25 genomes using two rounds of local blast. It’s obligatory to explain the feasibility of the results (Was this method adopted by others?), along with detailed settings (blastp or blastx or blast with conserved domains?)
2. The expression patterns of those genes were evaluated by using published data. However, it was hard to understand the feasibility of methods to normalize gene expressions and to determine the significance. For example, was this method widely used? How was the cutoff value chosen (0.1%)?
Results and conclusions:
1. There are reports about immune-responsive miRNAs in C. gigas under V. splendidus stimulation. Is there any correlation between expression patterns of those genes and miRNAs?
2. Was there any difference in the gene structure or expression pattern between miRNA-related genes from M. galloprovincialis and C. gigas?
3. The results and conclusions about digital expression analysis were unpersuasive and speculative. And less was discussed about potential influence, let alone experiment verification.
4. The date used here were too overwhelming, it might be more proper to focus on certain process with comparison between M. galloprovincialis and C. gigas.
Minor points
1. Line 55, citation “M.-J. Xu et al. 2014” could not be found in reference. The formats of citations in line 61 and 64 were different. Please check all the citations.
2. Line 85, “further 20 organisms”. A total of 25 genomes were used in the present paper and 2 of them were mentioned before. So, it should be “further 23 organisms”?
3. Line 106, the E-value should be “10-20”.
4. Line 124, it should be “2.995 G reads and 642 M reads”.

Reviewer 2 ·

Basic reporting

The article has a good format but is in parts too excessive as - as mentioned before - the rational for e.g. showing the figures 1-5 is not clear enough. The manuscript is well structure in material & methods, results and discussion, but e.g. for the part “Phylogenetic relationships of the key miRNA biogenesis proteins”, I am missing a section in material and methods. Parts of the text lack appropriate expression and should be seen by a native English speaker (L7: “mostly present in eukaryotes”, L28: “recognized as distinct arms”, L45 “Even if DROSHA and DGCR8”, L60 “To the date of this work”, L62 “Now-a-days”, L144 “It is known”, L157 “It is know”, L297 “As regards oyster”). Some scientific expressions are incorrect, too (L166 “Lophotrocozoans” instead of Lophotrochozoa, L209 “Gastropods” instead of gastropods, L234 “Arthropods” instead of arthropods, L293 “trocophora” instead of trochophora, L302 “Ostreid Herpesvirus” instead of ostreid or Ostreid herepesvirus, L318 “Lophotrochozoans” instead of lophotrochozoans). The absence of a reference to Zhou et al 2014 is a serious shortfall of the work and requires inclusion and accordingly adaptation of the manuscript (http://journals.plos.org/plosone/article?id=10.1371/journal.pone.0088397#pone.0088397.s001). Accordingly, I think it could be advisable to briefly review the status of miRNA complements in (lower) invertebrates (Flatworms: Fromm et al MBE 2014; Annelids: Helm et al MPE 2012 and miRNA in phylogenetic studies: Tarver et al MBE 2014 and others)

Experimental design

This is the weakest part of the manuscript. While the analyses are carried out correctly and in a reasonable way, it remains unclear what the researches want to achieve. While simply describing the sequences of the miRNA processing machinery is a reasonable endeavor, the comparative analyses lack crucial species like flatworms, where a lot is known and sequences might be annotated already. See Dunn et al for details (and update nomenclature! http://www.annualreviews.org/doi/pdf/10.1146/annurev-ecolsys-120213-091627). Once other relevant groups like flatworms are included I would like to ask the authors to use little more time to discuss the results of their “phylogenetic” analyses (especially results presented in Fig 3: blue line in green area, ID of uncolored clades between yellow and pink?
Similarly, I would strongly encourage the authors to decide either to put more work into the interpretation of their digital expression exercise, or leave it out. To me it is confusing and too much scratching of the surface, really.

Validity of the findings

The absence of a clear question influences the whole study and therefore the current conclusion of the manuscript is very weak. If the authors focus and strengthen the research question, a stronger conclusion will be possible, too. Although too much speculation is clearly not welcome, the authors should try to show potential paths their findings might have opened.

Additional comments

While the study is certainly interesting, it suffers from the absences of a clearly formulated research question and an incomplete taxon sampling for the comparative analyses. By sharpening the focus, formulating a clearer research question and by including additional available organisms, the manuscript will be suitable for publication.

·

Basic reporting

This paper takes an in silico approach to identify genes associated with miRNA biogenesis in marine bivalves, specifically Crassostrea gigas and Mytilus galloprovincialis. In total there is the report of key sequences associated with miRNA biogenesis in the genomes of 25 organisms and 31 bivalve transcriptomes. There is limited information on short RNAs in bivalves and this work can provided an important starting point for more targeted research on miRNA function in this taxa. Primary aspects of the work that need to be addressed include correcting English grammar (including title), a more complete explanation of the methods and data used in analysis (in manner that the analysis could be reproduced), and attention to overall organizational structure of the manuscript. I will elaborate on these aspects and indicate points that need attention below.

The entire work should be reviewed with attention to grammar. I will point out some issues but not all. One is the title itself.

"Mostly present in eukaryotes" is awkward phrasing, and somewhat confusing as introduction mentions work in bacteria.

minor: no need to say "cell" gene expression-

"uploaded in the RISC complex" is awkward phrasing

line 42 - should it be pre-miRNA?

line 45 "Even if DROSHA and DGCR..." this is the first time these proteins are mentioned and there should better defined. This could be done with just improved sentence structure.

Given the entire is work in on miRNA biogenesis, I suggest improving this paragraph (starting at line 42) from a grammar perspective. A diagram showing the major proteins would certainly improve the manuscript. This could be included in the results section where the authors illustrate the proteins that they have identified as part of their in silico analysis.

line 56 "Besides, " is awkward phrasing, there are several places where sentences start with unnecessary wording.

Experimental design

As in silico work, I include here specific comments regarding methods. Generally speaking, more information needs to be provided, and all results made available.

line 68 - the 8 specific proteins should be indicated. For table 1- Accession number and references should be provided for each genome and transcriptome used.

Line 72- I would say it is debatable whether the Mg WGS project results could be considered a genome draft.

Line 73 - For the Mg transcriptome, the data used should all be published data. If there is no way to publish this data it should be removed from the assembly.

Line 79- specific arguments (parameters) for Trinity assembly and Transdecoder prediction should be provided as well as the version.

Line 76- Was any quality assessment done to the sequences prior to assembly?

Line 79 - "Oyster mRNAs and predicted proteins were retrieved from genome annotations and manually updated using transcriptome assembly obtained from the available Illumina reads." This sentence does not explain what was done. Provide reference for annotations, then please describe how they were manually updated. I would recommend using what is available @ http://metazoa.ensembl.org/Crassostrea_gigas/Info/Index

Line 81 - at least list number of bivalve species and be specific on trinity assembly conditions.

Line 82 - It is good that "quality" was assessed for each assembly, but this requires a better description. Also- each transcriptome should be made available as part of this publication. Currently reads as "we retrieved and aligned the sequences of conserved proteins, like actin and El1α, with the reference deposited at NCBI, obtaining a perfect match". Does this mean that for each of the species there is a full length sequence of actin and El1a - and for every transcriptome assembled there was a 100% match?

Line 85. Need to be explicit about which protein sequences were previously published and which ones the authors generated. Were these assessed?

Line 90- HMMer parameters (and version) need to be specified.

Line 92-94- "Similarity searches were performed for the species listed in Table 1 by local BLAST, using human, Mg or Cg sequences as first queries. The top-ranking blast hits were used again as query sequences to identify candidate homologue genes for each species." needs clarification. What blast parameters were used? and How many top ranking blast hits were used for each species. And they were used as query sequences compared to what database?

Line 101- It is not clear what distinguishes this approach to the one described in paragraph before.

Line 107 - " the resulting contigs and singletons were manually combined using the corresponding transcripts as backbone." This needs to more precisely define - what is manually combined?

Line 109 - What program was used for RNA-seq mapping?

Line 125 - How were sequences quality trimmed - what were the settings

Line 130 - "A first normalization step was carried out considering the total mapped reads." please explain

Validity of the findings

See above section with regard to explaining the precise approaches used. Data on which conclusions based need to be provided.

Additional comments

Other comments
Line 140 - " identified other 21 proteins known to interact during the miRNA maturation or in consequent RNAi " - State in the methods how this was determined.

Much of the results text is material for introduction of discussion, describing what proteins. It would seem fundamental results would be a list of Mg and Cg proteins that match (including the ability of the reader to obtain these sequences - ) as well as some quantitative value of similarity (ie evalue). Table 3 does appear to do this, though lacking similarity. Also in my experience Genbank does not allow in silico derived transcript submission (ie third-party). As of review these accession numbers are not accessible.


Line 130 What does it mean to update the annotation?

Table 1. The homologs found, particularly for bivales need to be available to the reader.

Line 225- given the admittedly variable nature of datasets used I see limited value in any phylogenetic analysis.

Line 253 - one example where authors include methods in the results section.

Full gene sequences need to be made available.

Line 308- Seems odd that 5 publication on bivalve miRNA datasets are only first mentioned in discussion?


Figure 4 - Information on how to obtain sequences should be provided - accession numbers.

Figure 6 - This figure has limited value in it current form. Possibly limit to 8 proteins discussed in manuscript.


Supplemental file 2 - this should refer to specific sequence ids, and in final format not be a docx file, likely an image format.

Supplemental file 3. Mg expression should also include ID, annotation for each sequence should be quantified (ie evalue). Some description of library? column titles should be provided.

---

## Round 0.2 · Minor Revisions

This is a much improved manuscript over the original submission. There are still some areas that need improvement. For example as outlined by Reviewer 2 you have introduced a mistake into your manuscript that needs to be fixed throughout. The main area apart from this mistake that needs due care and attention is grammar and phrasing (throughout the manuscript). Please read all three excellent reviews carefully and make corrections accordingly.

Reviewer 1 ·

Basic reporting

No comments

Experimental design

No comments

Validity of the findings

No comments

Additional comments

The manuscript proposed a glance on the phylogeny and expression of genes involved in the miRNA biogenesis in M. galloprovincialis and other bivalves, such as C. gigas. The authors have made substantial improvements with clearer conclusion. However, there is still room for the improvement of the MS, especially the language. I personally encourage the authors to copyedit the paper to improve the style of written English.

Reviewer 2 ·

Basic reporting

see below

Experimental design

see below

Validity of the findings

see below

Additional comments

While the overall impression is better now, the manuscript still lacks some consitency and solidity.

You mention Ectocarpus paper in the introduction and cite the wrong Tarver et al. while you should cite: (1)
You mention miRBase but not MirGeneDB and the corresponding Fromm et al paper that summarizes the doubts about miRBase correctness in miRNA counts (2). A crucial paper you should have.
More importantly you introduced a mistake that requires editing of several parts of your manuscript and at least the phylogenetic analyses. You introduced, as I proposed, flatworms into your analysis but you count only 1 DICER gene. Please study Gao et al with more care and change accordingly (3). L 237 needs the Maxwell reference (4). Also I believe Figure 3 needs a little more care, too. Why do you show only uncommented trees while you summarize in Figure 4 so nicely? Please summarize the differences between the genetrees and discuss them.


1. Tarver JE, et al. (2015) microRNAs and the evolution of complex multicellularity: identification of a large, diverse complement of microRNAs in the brown alga Ectocarpus. Nucleic acids research 43(13):6384-6398.
2. Fromm B, et al. (2015) A Uniform System for the Annotation of Vertebrate microRNA Genes and the Evolution of the Human microRNAome. Annual review of genetics 49(1):null.
3. Gao Z, Wang M, Blair D, Zheng Y, & Dou Y (2014) Phylogenetic analysis of the endoribonuclease Dicer family. PloS one 9(4):e95350.
4. Maxwell EK, Ryan JF, Schnitzler CE, Browne WE, & Baxevanis AD (2012) MicroRNAs and essential components of the microRNA processing machinery are not encoded in the genome of the ctenophore Mnemiopsis leidyi. BMC genomics 13:714.

·

Basic reporting

Authors addressed prior comments

Experimental design

Authors addressed prior comments

Validity of the findings

Authors addressed prior comments

Additional comments

I am satisfied with the revisions made and responses to the previous reviews. The authors have significantly improved the manuscript. There are a few grammar issues and the Figures / Tables / Supp files do not match what is referred to in text and should be double-checked. I have attached the manuscript with suggested improvements in grammar, and a few minor comments.

---

## Round 0.3 · accepted · Accept

Thank you for incorporating suggestions raised in the previous review. The manuscript is now acceptable for publication, however, please ensure that you address the minor but important comment raised by Reviewer 2.

Reviewer 1 ·

Basic reporting

No comments.

Experimental design

No comments.

Validity of the findings

No comments.

Reviewer 2 ·

Basic reporting

P.2 L 28: There are currently 523 genes and NOT 532 in MirGeneDB

Experimental design

No Comments

Validity of the findings

No Comments